# AutoLoRA: Automatic LoRA Retrieval and Fine-Grained Gated Fusion for Text-to-Image Generation

## Abstract

Despite remarkable progress in photorealistic image generation with large-scale diffusion models such as FLUX and Stable Diffusion v3, the fragmented ecosystem of community-developed LoRA adapters and the difficulty of systematically integrating them into foundational models hinder their practical deployment. Their widespread adoption faces three pressing challenges: sparse metadata annotation, the requirement for zero-shot adaptation, and suboptimal strategies for multi-LoRA fusion. To address these challenges, we propose a framework that unifies community-developed LoRA adapters through semantic retrieval and dynamic fusion, effectively functioning as an ecosystem integrator. The framework consists of two key components: (1) a weight encoding-based retriever that aligns LoRA parameter matrices with text prompts in a shared semantic space, thereby eliminating the need for original training data, and (2) a fine-grained gated fusion mechanism that computes context-specific fusion weights across network layers and timesteps, enabling the optimal integration of multiple LoRAs during generation. Experiments demonstrate that our approach significantly outperforms strong baselines. Critically, AutoLoRA maintains high generation fidelity and aesthetic quality when fusing up to three distinct LoRAs, a challenging scenario where prior methods often suffer from catastrophic interference. Our framework not only improves automated aesthetic scores by substantial margins but also establishes a practical bridge between the community-driven proliferation of LoRA modules and the deployment requirements of large-scale diffusion systems.

## 1 Introduction

The proliferation of open-source model adapters has created a new kind of knowledge repository, yet we lack the fundamental tools to directly read and interpret these models from their weights. This paper takes a first step toward what we call model-based semantics, where LoRA adapters are no longer regarded as opaque parameter deltas but as carriers of reusable semantic capabilities. The rapid advancement of large-scale diffusion models such as FLUX and Stable Diffusion v3 has yielded astonishing capabilities in generating photorealistic and diverse imagery (Rombach et al., 2022; Esser et al., 2024; Labs, 2024). To support efficient customization, the community has widely adopted LoRA (Hu et al., 2022), a cost-effective fine-tuning technique that enables personalization with minimal parameter overhead. Thousands of LoRA adapters have already emerged in open-source communities such as ModelScope, Hugging Face, and Civitai, driving community growth and catalyzing innovation in AI-generated art. However, these LoRAs are scattered and isolated, often lacking standardized metadata or descriptive documentation, hindering discovery, retrieval, and integration of relevant adapters, making harnessing of this vast ecosystem a significant challenge.

A practical solution is to construct a LoRA retrieval and fusion system capable of fully utilizing community-contributed adapters to improve text-to-image generation. However, retrieving and fusing relevant LoRAs poses unique challenges compared to other retrieval-augmented generation (RAG) systems (Lewis et al., 2020; Gao et al., 2023). First, effective retrieval requires embedding representations of LoRAs. Existing approaches such as PHATGOOSE (Muqeeth et al., 2024a), which learns an embedding vector jointly with LoRA parameters during LoRA training, and SemLA (Qorbani et al., 2025), which derives embeddings from the image data used to train LoRAs, both

assume access to the original training process or dataset. In practice, open-source LoRAs rarely include training data or detailed documentation, making document-based or data-dependent retrieval infeasible. Furthermore, the LoRA ecosystem is continuously expanding, requiring retrieval systems to support zero-shot adaptation for newly released adapters. Second, multi-LoRA fusion remains a major challenge. Naïve linear combinations or distillation typically suppress the performance of individual LoRAs, and performance deteriorates as the number of adapters increases. Recent works have explored mixture-of-experts (MoE) approaches (Wu et al., 2024; Zhao et al., 2024), but these methods rely on a fixed expert pool, which fundamentally limits their scalability. Such designs are incompatible with the open-world nature of community-driven LoRA ecosystems, where the set of available adapters grows dynamically and cannot be predetermined.

To address these challenges, we propose AutoLoRA, a framework for LoRA retrieval and dynamic aggregation that leverages the collective capabilities of open-source adapters to improve text-to-image generation. The framework consists of two key components: (1) Weight encoding-based LoRA retriever: Motivated by the success of CLIP in multimodal representation learning, we design a weight encoder that maps both LoRA weights and text prompts into a shared feature space via contrastive learning. Since LoRA weights differ significantly from natural language or images, conventional encoders are inadequate. We therefore tokenize each layer's weight matrix, embed tokens through trainable parameters, and use transformer blocks to produce a compact representation of the entire LoRA. (2) Fine-grained gated fusion mechanism: Inspired by recent findings that LoRAs exhibit layer- and timestep-specific effects within diffusion models (Ouyang et al., 2025), we argue that a truly effective fusion mechanism must operate at this fine-grained level. To this end, AutoLoRA employs learnable gating modules at each linear layer, which dynamically condition on both the hidden states of the base model and LoRA-modulated features across diffusion timesteps. Unlike naïve fusion schemes that assign fixed weights, our mechanism flexibly adjusts contributions of each LoRA across feature dimensions and timesteps, enabling robust integration of multiple adapters and maximizing generative performance.

To evaluate AutoLoRA, we collected 162 FLUX LoRAs from open-source platforms, covering diverse themes, tasks, and architectural variants. Although these community-contributed adapters lack accessible training data or detailed documentation, most include a few sample renderings. We leverage Qwen-VL-Max to caption these renderings, constructing paired training and evaluation sets. Experiments show that only a small number of renderings per LoRA suffice to train the weight encoder and gated fusion module, and that retrieving and aggregating LoRAs with AutoLoRA consistently improves text-to-image generation. In summary, AutoLoRA introduces a weight-encoding–based retrieval model and a dynamic gated fusion mechanism, enabling semantic-driven retrieval and harmonious aggregation of multiple LoRAs. Our approach not only significantly improves automated aesthetic scores but also establishes a practical bridge between the community-driven proliferation of LoRA modules and their deployment in large-scale diffusion systems, taking a first step toward interpreting the semantic functionality encoded in model weights. An anonymous code repository is available at https://anonymous.4open.science/r/AutoLoRA-6759.

## 2 RELATED WORKS

### 2.1 ADAPTER RETRIEVAL

As adapter technology matures, the proliferation of adapters within the community has prompted growing research interest in retrieving task-specific adapters. A straightforward approach involves leveraging traditional MoE techniques (Lepikhin et al., 2021; Fedus et al., 2022) for adapter retrieval, where methods such as SMEAR (Muqeeth et al., 2024b) and MoLE (Wu et al., 2024) introduce an additional routing module during training. This module dynamically directs input tokens to different adapter experts based on their semantic content. The primary limitation of MoE-based approaches is that the number of experts remains fixed and typically limited, leading to insufficient flexibility and scalability. SemLA (Qorbani et al., 2025) proposes a training-free framework for image segmentation that retrieves adapters by measuring the similarity between the input image and the adapter training dataset. PHATGOOSE (Muqeeth et al., 2024a), in contrast, learns a LoRA signature vector from the training dataset during LoRA model training and subsequently retrieves adapters by computing the similarity between input tokens and the signature vector, thereby enhancing LLM performance across diverse tasks and improving adaptability in zero-shot scenarios.

RAMoLE (Zhao et al., 2024) trains a sentence-embedding model via instruction fine-tuning and retrieves task-specific LoRA models in LLMs by analyzing the similarity of sentence embeddings derived from input text. While effective, these methods are mostly tailored for LLMs and require adapter training data, limiting their generalizability to image generation.

## 2.2 ADAPTER FUSION

In the field of image generation, integrating personalized Adapters has long been a significant challenge, prompting numerous studies to be proposed (Dong et al., 2024; Yang et al., 2024; Gu et al., 2023). ZipLoRA (Shah et al., 2024) proposes a straightforward method to learn scaling coefficients that render the columns of two adapters' weight matrices nearly orthogonal, thereby preventing interference when combining the adapters. Inspired by recent advances in Mixture of Experts (MoE) techniques, MoLE (Zhao et al., 2024) treats each adapter as an expert module and trains a gating mechanism within every feed-forward network layer to dynamically modulate the contribution of each adapter across different model layers. Unlike methods that combine different models through integration, LoRACLR (Simsar et al., 2025) distills knowledge from multiple target-concept adapter models across various image generation frameworks into a single adapter, enabling accurate generation of images depicting multiple concepts simultaneously. the recent K-LoRA (Ouyang et al., 2025) propose a training-free approach that calculates the top-k elements from each target model within every attention layer to dynamically determine which model should be activated at each step. Similarly, DARE (Yu et al., 2024) employs a training-free strategy inspired by stochastic dropout, randomly discarding incremental parameters from different models according to specific policies to mitigate conflicts during model fusion. Although effective in controlled settings, these methods typically focus on fusing only a small number of fixed LoRAs. They still encounter critical limitations—including poor subject consistency and high training complexity—making them impractical for scenarios where we dynamically retrieve arbitrary numbers of diverse LoRAs for each prompt.

# 3 AUTOLORA FRAMEWORK

## 3.1 WEIGHT ENCODING-BASE LORA RETRIEVER

Our goal is that the user inputs a text prompt, and the retriever can recall k LoRAs associated with the text prompt from the LoRA pool, and the number of LoRAs in the pool of LoRAs is incrementally updated. To achieve this goal, we use a CLIP (Radford et al., 2021) model architecture which contain a text encoder and a LoRA encoder, they can encode text and LoRA into an embedding respectively, and then calculate the similarity between the embeddings to complete the retrieval. The text encoder we can use pre-trained model, but how do we design a LoRA encoder that can input a LoRA weight parameter and output an embedding? As we all know, LoRA introduce low-rank matrices to each linear layer. Specifically, for a linear with weight $W_0 \in \mathbb{R}^{d \times k}$, LoRA augment it with two low-rank matrics: $W_0 + \Delta W = W_0 + BA$, where $B \in \mathbb{R}^{d \times r}, A \in \mathbb{R}^{r \times k}$ and $r \ll \min(d, k)$. Further, we denote a LoRA as $\Delta W = (B, A)$, where $B = \{B_1, \cdots, B_m\}$ and $A = \{A_1, \cdots, A_m\}$ denote LoRA is applied to $m$ linear layers of the original model. We conceptualize this encoding process as "probing" the LoRA layer to reveal its functional semantics. For each layer $i$, we introduce a trainable probe vector $q_i \in \mathbb{R}^{1 \times d_{in}}$, where $d_{in}$ is the input dimension of the linear layer (corresponds to $k$ in your notation). This probe acts as a canonical input signal. The LoRA layer's response to this probe is given by the transformation it applies:

$$\text{response}_i = q_i (B_i A_i)^T = q_i A_i^T B_i^T. \tag{1}$$

This response, a vector in $\mathbb{R}^{1 \times d_{out}}$, captures how the LoRA layer alters signals along the direction of the probe. To map this response into the shared embedding space of dimension $o$, we apply a trainable projection matrix $\hat{W}_i \in \mathbb{R}^{d_{out} \times o}$. Thus, the token embedding $v_i$ for the $i$-th layer is:

$$v_i = (q_i A_i^T B_i^T) \hat{W}_i. \tag{2}$$

By making $q_i$ and $\hat{W}_i$ trainable and layer-specific, the model learns to automatically discover the most informative "virtual inputs" and projection schemes to distill the essential semantics of each LoRA layer. This process is denoted as $v = \text{Embedder}((B,A))$, the sequence of these embeddings $v = [v_1, \ldots, v_m]$ is then fed into a Transformer encoder to produce the global LoRA representation:

$$e = \text{Encoder}(v), \tag{3}$$

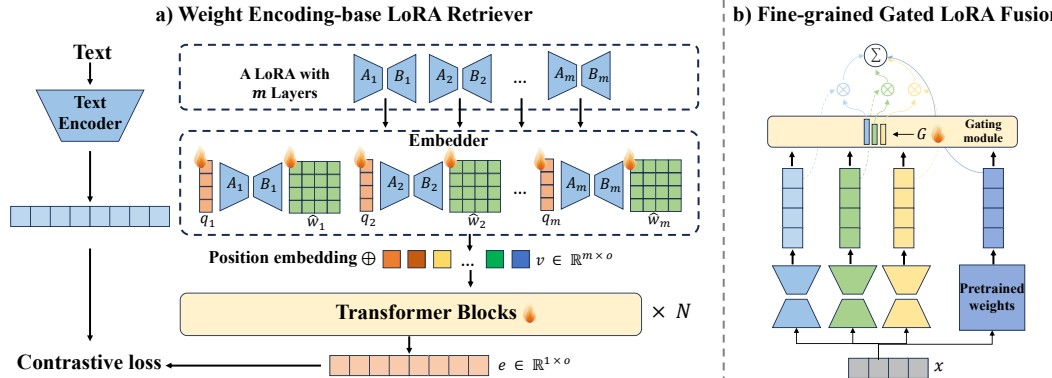

Figure 1: Illustration of the AutoLoRA framework, which consists of two main components: (a) Weight Encoding–Based Retriever, where a LoRA weight encoder maps both LoRA parameters and textual prompts into a shared embedding space through contrastive learning, enabling semantic-driven retrieval; and (b) Fine-Grained Gated Fusion, which employs learnable gating modules to dynamically regulate LoRA contributions across different dimensions, thereby facilitating effective integration and mitigating conflicts among multiple LoRAs.

where $e \in \mathbb{R}^o$ is the final representation embedding of LoRA and Encoder$(\cdot)$ represents the feature encoder composed of $N$ standard Transform Blocks. The overall model structure of LoRA Encoder is shown in the Figure 1. We compare different LoRA Encoders, validating the effectiveness of our probe vector $q_i$ (see Appendix G.5 for details).

**Training Object.** We use contrastive learning for training. First, we use VLM to convert each LoRA rendering into text, and use these texts as the labels corresponding to each LoRA. In this way, we can get a data set $\mathcal{T} = \{(x_{1,1}, (B, A)_1), \cdots, (x_{j,t}, (B, A)_j)\}$, where each LoRA corresponds to at least one text description. We use the pre-trained CLIP text encoder to encode text: $t = \text{CLIP}_{text}(x)$, and freeze the parameters of the text encoder during training, only update the parameters of the LoRA encoder, and take $N$ data from $\mathcal{T}$ for training each iteration. The loss function is as follows:

$$\mathcal{L} = \sum_{i=1}^{N} \left( -\log \frac{\exp(e_i^\top t_i)}{\sum_{j=1}^{N} \exp(e_i^\top t_j)} - \log \frac{\exp(e_i^\top t_i)}{\sum_{j=1}^{N} \exp(e_j^\top t_i)} \right) \tag{4}$$

In the LoRA retrieval stage, given a LoRA set $\Phi$ and an input prompt $x$, the $top - k$ LoRAs are retrieved according to the cosine similarity:

$$s = cos(\text{Encoder}(\text{Embedder}((B,A))), \text{CLIP}_{text}(x)). \tag{5}$$

This process can be expressed as:

$$\Phi_k = \text{TopK}\{s((B, A)_j, x), (B, A)_j \in \Phi\}. \tag{6}$$

### 3.2 FINE-GRAINED DYNAMIC GATED LORA FUSION

After retrieving $k$ relevant LoRAs, the next thing we need to do is to integrate these LoRAs into the diffusion model. A straightforward idea is to use the MoE method (Wu et al., 2024) to train an additional router in each layer and assign different weights to each LoRA. However, the traditional MoE method requires a fixed number of LoRAs during training, and can only assign weights to these fixed LoRAs during inference. It cannot be applied to the scenario of dynamically selecting LoRAs from the LoRA pool for fusion. To address this challenge, we propose a Fine-grained dynamic gating LoRA fusion mechanism, which utilizes a learnable gating module in the linear layer to perceive the hidden state features of each intermediate layer of the original model and LoRA during the diffusion process, and dynamically calculates the LoRA weights of different dimensions.

Formally, consider top-k retrieved LoRAs in a linear layer. The output of original model output $\mathbf{x} \in \mathbb{R}^{l \times d}$ and a collection of LoRA outputs $\mathbf{L} = [\mathbf{l}_1, \mathbf{l}_2, \ldots, \mathbf{l}_k] \in \mathbb{R}^{k \times l \times d}$, the module first applies normalization to both inputs to eliminate scale discrepancies and highlight critical features:

$$\hat{\mathbf{x}} = \text{LayerNorm}(\mathbf{x}), \quad \hat{\mathbf{L}} = \text{LayerNorm}(\mathbf{L}). \tag{7}$$

Subsequently, our gated mechanism computes dimension-specific contribution weights for each LoRA through the synergistic operation of two specialized gate components:

$$\mathbf{G} = \sigma \left( \hat{\mathbf{L}} \odot \mathbf{w}_l + \hat{\mathbf{x}} \odot \hat{\mathbf{L}} \odot \mathbf{w}_c + \mathbf{b} \right), \tag{8}$$

Here, $\mathbf{w}_l, \mathbf{w}_c, \mathbf{b} \in \mathbb{R}^d$ are learnable weight and bias vectors. They are broadcasted to match the dimensions of $\hat{\mathbf{L}} \in \mathbb{R}^{k \times l \times d}$ for the element-wise operations. The term $\hat{\mathbf{L}} \odot \mathbf{w}_l$ allows the gate to weigh LoRA contributions based on their own output features, while the interaction term $\hat{\mathbf{x}} \odot \hat{\mathbf{L}} \odot \mathbf{w}_c$ makes the gating decision conditional on the interplay between the base model's state and the LoRA's modification (See Appendix G.1 for gate ablation results). And $\mathbf{b} \in \mathbb{R}^d$ is a learnable bias term, $\sigma$ denotes the sigmoid activation function, and $\odot$ represents element-wise multiplication. The resulting gating matrix $\mathbf{G} \in \mathbb{R}^{k \times l \times d}$ contains dynamic weights $g_{i,j,d}$ that determine the contribution strength of the $i$-th LoRA to the $d$-th feature dimension of the $j$-th token representation, with values determined through the collaborative decision-making of the two gate components. Finally, the module integrates the original output with the weighted LoRA outputs through amplitude calibration:

$$\mathbf{x}' = \mathbf{x} + \sum_{i=1}^{k} \left( \mathbf{w}_o \odot \mathbf{g}_i \odot \mathbf{l}_i \right), \tag{9}$$

where $\mathbf{w}_o \in \mathbb{R}^d$ is a learnable Fusion-Scaling Parameter specific to each layer, which further ensures numerical stability during integration and $\mathbf{g}_i$ represents the $i$-th row of $\mathbf{G}$.

**Global LoRA.** Inspired by recent MoE approaches (Dai et al., 2024) that leverage shared experts to capture and integrate general knowledge across diverse contexts, we introduce a Global LoRA in our fusion framework. Unlike a standalone adapter, this Global LoRA is synthesized by summing the weight matrices of the target LoRAs and then decomposing the aggregated matrix into two low-rank components via matrix decomposition. From a matrix-theoretic perspective, the summation $\sum_{i=1}^{n} B_i A_i$ corresponds to combining multiple low-rank updates. Performing a low-rank decomposition (e.g., SVD) on this sum amounts to identifying the principal directions that best capture the subspace spanned by these updates. In this way, $B_g, A_g$ represent the optimal low-rank approximation of the aggregated update directions, yielding a compact representation of their common knowledge. The global LoRA is seamlessly integrated into both training and generation phases:

$$B_g, A_g = \mathcal{D}_{r_g} \left( \sum_{i=1}^{n} B_i A_i \right), \tag{10}$$

where $\mathcal{D}_{r_g}(\cdot)$ denotes the matrix decomposition algorithm. We adopt singular value decomposition (SVD), and for efficiency, we employ PyTorch's approximate SVD implementation by restricting the decomposition rank to a small value . The rank $r_g$ of the global LoRA acts as a hyperparameter controlling the capacity of the shared representation. A small rank enforces a strong bottleneck, capturing only the most dominant shared semantics, while a larger rank allows for more nuanced shared features at the risk of overfitting or capturing noise. We empirically found $r_g = 4$ to be a sweet spot (see Appendix G.2 and G.3 for details).

**Training Object.** We use a strategy called Interference-Resistant Training to train fusion module. During each training iteration, we randomly sample two LoRAs from the pool: one designated as the target LoRA $L_i$ and the other as the interference LoRA $L_j$. While both LoRAs are simultaneously active in the forward pass, the training signal is exclusively derived from the target LoRA's image-text pairs. This asymmetric supervision forces the fusion module to learn discriminative gating behavior—effectively amplifying relevant features while suppressing interference—despite the concurrent presence of both adapters. Intuitively, it provides the gating module with a direct, reconstruction-error-based supervisory signal: when a LoRA's contribution fails to improve alignment with the target, its gating weight is suppressed, while contributions consistent with the target are amplified. The optimization objective employs the flow matching loss:

$$L = \mathbb{E} \left[ \left\| V_{\hat{\theta}} \left( x_t, c, t, L_i, L_j, L_g \right) - (x_1 - x_0) \right\|^2 \right], \tag{11}$$

where $V_{\hat{\theta}}$ is the diffusion model with the LoRA fusion mechanism, $L_g$ is the global LoRA constructed by $L_i$ and $L_j$, $x_t$ is the target image in latent space, $x_0 \sim N(0,1)$ is the noise, $c$ is the text condition, and $t \sim \mu(0,1)$ is the timestep. We compared different training strategies for the fusion module and validated that our Interference-Resistant approach helps resolve conflicts when combining multiple LoRAs (see Appendix G.4 for details).

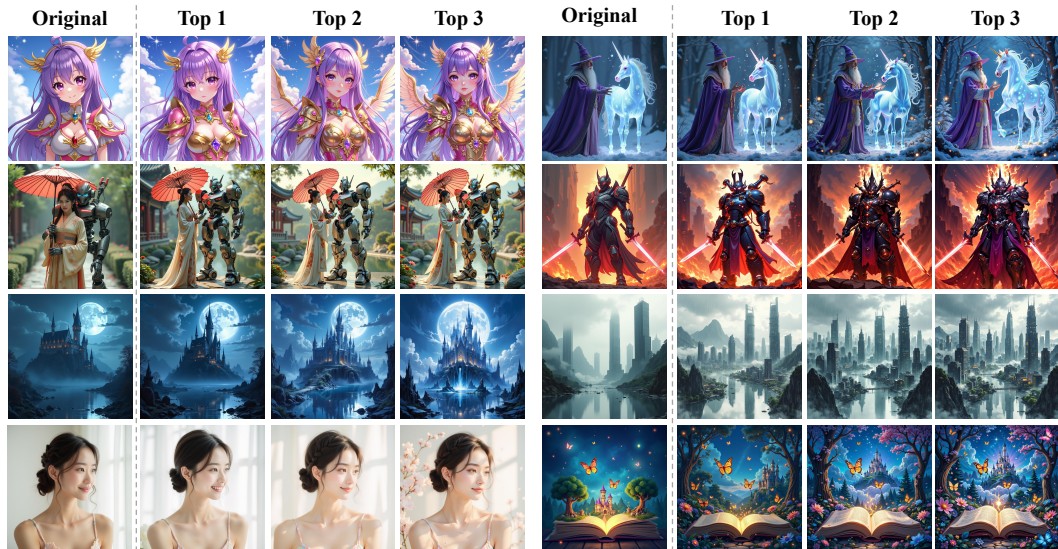

| Original | Top 1 | Top 2 | Top 3 | Original | Top 1 | Top 2 | Top 3 |

Figure 2: Qualitative comparison between AutoLoRA and the FLUX.1-dev. Top-1, Top-2, and Top-3 denote images generated by retrieving and fusing 1, 2, and 3 LoRAs, respectively. AutoLoRA enriches visual details, refines artistic characteristics, and enhances overall aesthetic quality.

## 4 EXPERIMENTS

### 4.1 RETRIEVAL-AUGMENTED FUSION FOR IMAGE GENERATION

**Experimental Setup.** We collected 162 diverse FLUX.1-dev LoRAs from multiple open-source platforms, including only their weight parameters and a few reference images (1–5 per LoRA). Descriptive captions for each reference image were generated using Qwen-VL-Max. For retrieval experiments, we randomly sampled 1–3 LoRAs from the pool and synthesized prompts from their reference images. Qwen-VL-Max produced prompts capturing both semantic and stylistic features, resulting in a synthetic benchmark of 900 prompts. To test generalizability, we additionally sampled 1,000 real user-generated prompts from DiffusionDB (Wang et al., 2023) and rewrote them with an LLM, introducing further diversity and complexity. Detailed procedures for dataset construction, prompt synthesis, and model training are provided in Appendix B.

We compared AutoLoRA with a baseline that retrieves LoRAs by cosine similarity between CLIP-encoded prompts and LoRA reference images; both methods integrate the retrieved LoRAs using the gated fusion mechanism during generation. Evaluation metrics include image aesthetic quality measured by MPS (Zhang et al., 2024), HPS (Wu et al., 2023), and Aesthetic (Schuhmann et al., 2022), as well as text–image alignment assessed with VQAScore (Lin et al., 2024). For each prompt, we retrieved the top-1, top-2, and top-3 LoRAs to generate images. Appendix E details AutoLoRA's computational and memory costs, which remain acceptable.

**Result Analysis.** Table 1 presents the quantitative results of AutoLoRA on both datasets. Compared to a text–image similarity retrieval baseline, AutoLoRA achieves consistent improvements across all three aesthetic metrics as well as text–image alignment. This demonstrates that AutoLoRA is capable of retrieving LoRAs relevant to the input prompt; these retrieved LoRAs help the model better interpret and emphasize specific elements in the prompt, and the gated fusion mechanism further maximizes their utility—ultimately producing images of higher quality that more faithfully follow textual instructions. We further observe that while performance generally improves as more LoRAs are retrieved, the marginal gains diminish. This is because retrieval is similarity-based: the top-ranked LoRA, being most relevant to the prompt, provides the largest improvement, whereas subsequent LoRAs with lower similarity scores contribute progressively smaller benefits. It is also due to semantic-space crowding: when multiple LoRAs attempt to modify overlapping feature dimensions, even fine-grained gating mechanisms face increasing coordination difficulties, which grow nearly exponentially with the number of adapters. Figure 2 presents qualitative results. Compared

to outputs from FLUX.1-dev, integrating the retrieved LoRAs substantially improves visual fidelity, enriches structural detail, enhances stylistic coherence, and elevates the overall aesthetic quality of the generated images. We also provide a detailed analysis of representative failure cases in LoRA retrieval and fusion in Appendix F, highlighting the challenges of aligning retrieved adapters with nuanced user intent and integrating multiple semantically conflicting LoRAs.

Results on the DiffusionDB dataset show that AutoLoRA maintains effectiveness on out-of-distribution prompts, demonstrating the robustness and generalizability of our framework. In contrast, retrieval methods based solely on text–image similarity perform poorly on this dataset, as such a simple strategy struggles to cope with more complex scenarios. Figure 2 presents qualitative results. Compared to outputs from FLUX.1-dev, integrating the retrieved LoRAs substantially improves visual fidelity, enriches structural detail, enhances stylistic coherence, and elevates the overall aesthetic quality of the generated images. Appendix D provides results from human assessments, confirming that images generated by AutoLoRA better align with user intent and exhibit higher aesthetic quality compared to baseline methods.

Table 1: Quantitative results using AutoLoRA on synthetic prompt set and DiffusionDB

| Dataset | Method | | MPS(↑) | HPS(↑) | Aes.(↑) | VQA(↑) |
|---|---|---|---|---|---|---|
| Synthetic prompt set | FLUX.1 dev | | 17.294 | 0.324 | 6.302 | 0.916 |
| | Text–Image Similarity Retrieval | Top 1 | 17.485 | 0.326 | 6.292 | 0.919 |
| | | Top 2 | 17.590 | 0.330 | 6.321 | 0.921 |
| | | Top 3 | 17.593 | 0.329 | 6.334 | 0.920 |
| | AutoLoRA | Top 1 | 17.523 | 0.329 | 6.300 | 0.920 |
| | | Top 2 | 17.634 | 0.335 | 6.362 | 0.922 |
| | | Top 3 | **17.749** | **0.340** | **6.401** | **0.922** |
| DiffusionDB | FLUX.1 dev | | 17.887 | 0.315 | 6.425 | 0.849 |
| | Text–Image Similarity Retrieval | Top 1 | 17.729 | 0.316 | 6.454 | 0.855 |
| | | Top 2 | 17.722 | 0.316 | 6.465 | 0.855 |
| | | Top 3 | 17.593 | 0.315 | 6.467 | 0.852 |
| | AutoLoRA | Top 1 | 18.072 | 0.328 | 6.496 | 0.857 |
| | | Top 2 | 18.105 | 0.332 | **6.522** | 0.861 |
| | | Top 3 | **18.166** | **0.334** | 6.515 | **0.861** |

## 4.2 EFFECTIVENESS OF FINE-GRAINED GATED FUSION

**Experimental Setup.** Following K-LoRA (Ouyang et al., 2025), we first test fusion on 3 object and 8 style LoRAs, generating 10 images per object-style prompt using the template: "a {Object} in the {Style} style." We then compute CLIP (Radford et al., 2021) similarity between the generated images and both the object and style references. For the more general setting, we construct two separate test sets by randomly sampling 2 or 3 LoRAs from the candidate pool, with each set containing 300 prompts. For each sampled combination, a single descriptive prompt is generated from the cover images of all selected LoRAs using Qwen-VL-Max. This prompt consolidates visual information from multiple images, capturing the features of all involved LoRAs, and forms a diverse LoRA fusion test set. Performance is measured using two complementary metrics: (1) image aesthetic scores and text–image alignment, and (2) similarity between fused outputs—images generated by integrating multiple LoRAs—and the outputs of individual LoRAs, quantifying how well the fusion preserves the distinctive characteristics of each LoRA.

**Result Analysis.** From the results of Object and Style LoRA fusion in Table 2, it is evident that our fine-grained gated fusion method substantially outperforms alternative baselines in simultaneously preserving object fidelity and style consistency. Direct linear addition tends to prioritize stylistic attributes at the expense of object fidelity, while K-LoRA, which specifically amplifies the style LoRA during generation, thereby dominates style features and reduces overall balance. Tables 3 report the results for fusing randomly selected LoRAs. K-LoRA is tailored for Object-Style fusion and exhibits suboptimal performance in these generalized scenarios. Moreover, its effectiveness is highly dependent on the LoRA loading order, as it assumes the second loaded LoRA represents the style component and disproportionately emphasizes it. Direct linear addition and DARE, lacking any

Figure 3: Qualitative comparison of object-style LoRA fusion. Compared with other baseline methods, our fine-grained gating fusion mechanism can seamlessly integrate multiple LoRAs while both ensuring object consistency and effectively preserving style attributes.

Table 2: Quantitative results of object and style LoRA fusion compared to baselines.

| Method | Direct | K-LoRA | DARE | Ours |
|---|---|---|---|---|
| Obj Sim ($\uparrow$) | 0.728 | 0.639 | 0.732 | 0.742 |
| Style Sim ($\uparrow$) | 0.579 | 0.624 | 0.577 | 0.577 |

selective modulation mechanism, suffer from mutual suppression among LoRAs. Particularly when fusing three LoRAs, this interference leads to severe image degradation and feature entanglement, resulting in distorted object shapes, incoherent styles, or mixed visual artifacts. In contrast, our approach remains robust regardless of the number of LoRAs fused. By assigning dimension-wise gating weights from latent features, our mechanism amplifies relevant contributions and suppresses conflicts, preserving image quality. As shown in Figure 3, it integrates LoRAs seamlessly—for example, producing a cat in oil painting style rather than a cat against an oil-painted background

Table 3: Comparison of random Multiple LoRAs fusion. "Original" denotes image generation without LoRA, "$l_i$-Sim" represents similarity to the $i-th$ LoRA's generated image.

| Method | Dataset | MPS | HPS | VQA | $l_1$-Sim | $l_2$-Sim | Dataset | MPS | HPS | VQA | $l_1$-Sim | $l_2$-Sim | $l_3$-Sim |
|---|---|---|---|---|---|---|---|---|---|---|---|---|---|
| Original | | 18.28 | 0.33 | 0.91 | 0.82 | 0.82 | | 18.35 | 0.33 | 0.92 | 0.83 | 0.82 | **0.85** |
| Direct | | 17.19 | 0.31 | 0.89 | 0.85 | 0.84 | | 16.78 | 0.29 | 0.89 | 0.82 | 0.81 | 0.81 |
| K-LoRA | 2-LoRA | 17.40 | 0.31 | 0.90 | 0.84 | **0.88** | 3-LoRA | \ | \ | \ | \ | \ | \ |
| DARE | Fusion | 16.41 | 0.29 | 0.86 | 0.83 | 0.82 | Fusion | 14.88 | 0.25 | 0.81 | 0.77 | 0.76 | 0.81 |
| Ours | | **18.42** | **0.34** | **0.93** | **0.86** | 0.85 | | **18.61** | **0.34** | **0.93** | **0.86** | **0.85** | **0.85** |

## 4.3 EFFECTIVENESS OF LORA ENCODER

To validate the effectiveness of the LoRA Encoder, we selected 20 LoRAs from the dataset covering six distinct themes (Cyberpunk Style, Ghibli Style, Mecha, Pixel Style, Face Enhancement, and 3D Rendering) and computed pairwise embedding similarities for all possible pairs. As visualized in the left panel of Figure 4 using a heatmap, LoRAs belonging to the same theme are highlighted by bounding boxes, indicating strong intra-theme coherence. We further calculated the average intra-class and inter-class similarity for each theme, as summarized in the right panel of Figure 4. In all themes, intra-class similarity exceeds inter-class similarity, confirming that the LoRA Encoder successfully maps semantically related LoRAs into neighboring regions of the feature space, facil-

itating accurate and robust retrieval in practice. Overall, these results suggest that the embeddings produced by the LoRA Encoder are semantically meaningful and effective for downstream LoRA retrieval tasks, demonstrating that we have successfully built a tool capable of directly reading and interpreting the semantic content of model weights, breaking the black-box nature of parameterized models and making "model-as-semantic" a practical possibility.

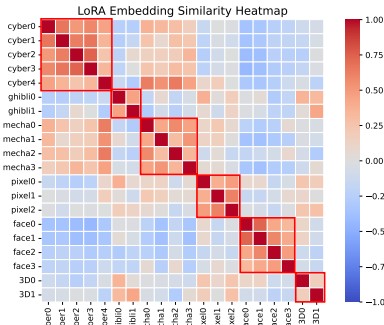

| Theme | Nums | Intra-class(↑) | Inter-class(↓) |
|---|---|---|---|
| Cyber | 5 | 0.492 | -0.043 |
| Ghibli | 2 | 0.434 | -0.019 |
| Mecha | 4 | 0.453 | 0.015 |
| Pixel | 3 | 0.600 | 0.042 |
| Face | 4 | 0.474 | -0.169 |
| 3D | 2 | 0.368 | 0.036 |
| Average | 20 | 0.471 | -0.023 |

Figure 4: Left: Heatmap of pairwise cosine similarities among 20 LoRAs, LoRAs from the same theme are grouped in square brackets. Darker cells denote higher similarity. Right: Average intra-class (within-theme) and inter-class (across-theme) similarities across six themes, showing that Lo-RAs within the same theme are more coherent than across themes.

## 4.4 GATING MODULE ANALYSIS

To verify that our gating module dynamically regulates LoRA contributions during generation, we visualize the gating values $G$ in an attention layer (blocks.18.attn.a_to_qkv). Using one Object LoRA and one Style LoRA, we generate images and track the top-512 summed values of each gating vector $g_i$ across timesteps (Fig. 5). The results show that the Object LoRA dominates at early stages, while the Style LoRA gains influence later, aligning with the diffusion process where global structure forms first

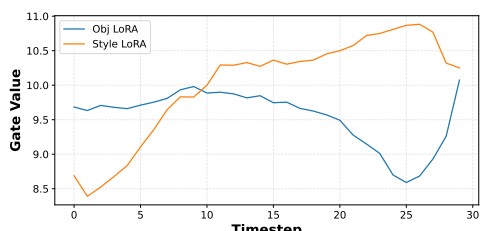

Figure 5: Gate values across timesteps.

and fine-grained style emerges later. This demonstrates that the gating module adaptively allocates weights to different LoRAs, maximizing their utility while mitigating conflicts.

## 5 CONCLUSION AND POTENTIALS

In this work, we introduce AutoLoRA, a unified framework that combines a weight-encoding retriever with a fine-grained gated fusion mechanism. AutoLoRA addresses zero-shot retrieval from sparse metadata and robustly fuses multiple, potentially conflicting adapters. Experiments show that it improves generative quality and fidelity while providing a scalable, data-efficient bridge between decentralized LoRA creation and centralized deployment in foundational models. Beyond practical gains, this work establishes a new paradigm of *model-based semantics*, viewing LoRAs as structured carriers of semantic information—styles, object concepts, or complex relations—rather than opaque parameter deltas. Our LoRA encoder maps raw weights to a coherent semantic space, enabling functional intent to be inferred directly from parameters and transforming unstructured weights into a queryable knowledge base. This semantic space opens rich avenues for future research. Model arithmetic could synthesize new capabilities—for example, 'Ghibli Style' - 'Anime Character' + 'Landscape' might produce a LoRA specialized for Ghibli-style landscapes. Semantic model editing allows subtle manipulation of embeddings—e.g., shifting a style toward 'more vintage' or a concept toward 'more abstract'—without retraining. Automatic capability discovery could cluster embeddings from large, unannotated LoRA collections to reveal emergent concepts and novel artistic or functional modules. This approach may reshape interactions with machine learning models, moving from monolithic training to modular, semantic composition, editing, and understanding.

## ETHICS STATEMENT

This work introduces AutoLoRA, a unified framework for retrieving and fusing community-contributed LoRA adapters to enhance text-to-image generation. Our focus is developing a weight-encoding-based retriever and a gated fusion mechanism. All experiments are conducted using publicly available, open-source LoRAs, without involving sensitive data, user information, or NSFW content. As such, this research does not pose specific ethical or safety risks and aims to promote scalable and responsible use of community-contributed LoRA adapters in foundational models.

## REPRODICIBILITY STATEMENT

We provide detailed experimental settings, model architectures, and implementation details in the appendix to ensure reproducibility. An anonymous code repository is available to facilitate verification and replication of our results. All datasets used in our experiments will be publicly released upon acceptance of the paper. These resources allow other researchers to reproduce our experiments and extend our AutoLoRA framework in a transparent and verifiable manner.

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

## A  USES OF LLMS

Large language models (LLMs) are used exclusively in this work for translation, text polishing, and grammar correction. All experimental design, data collection, model training, and analysis are performed entirely by the authors. LLMs do not generate, modify, or influence any experimental results or conclusions.

## B  IMPLEMENTATION DETAILS

### B.1  LORA ENCODER

The LoRA Encoder consists of two components: an Embedder and Transformer Blocks. The main trainable parameters in the Embedder are $q_i \in \mathbb{R}^k$ and $\hat{W}_i \in \mathbb{R}^{d \times o}$. The values of $k$ and $d$ depend on the corresponding LoRA layer, while the output dimension $o$ is fixed at 768. The Transformer component comprises 2 blocks with 12 attention heads, each of dimension 64, and a feed-forward network with a hidden size of 3072. The total parameter count of the encoder is approximately 112.9M. For the text encoder, we adopt the pretrained CLIP-ViT-Large-Patch14. Training is performed with a learning rate of $1 \times 10^{-4}$ using the AdamW optimizer and a batch size of 32. The model is trained for 5 hours on 4 × AMD ASPEED (rev 52) GPU.

### B.2  LORA FUSION MODULE

For each linear layer in the base model where a LoRA adapter is inserted, we augment it with a gating module. Each gate is a vector of dimension $d$, resulting in a total parameter count of only 27.3M for the entire gating module. Training is conducted with a learning rate of $1 \times 10^{-4}$ using the AdamW optimizer for 8,000 steps. The process takes approximately 7 hours on 4 × AMD ASPEED (rev 52) GPU.

## C  DATASET CONSTRUCTION DETAILS

### C.1  LORA SELECTION STRATEGY

We collected LoRA models primarily from two open-source platforms: ModelScope and Civitai. Starting from the top-ranked models by download count, we initially obtained over 1,100 LoRAs. Given the varying quality of community-contributed LoRAs, and the presence of some with inappropriate content, we applied further filtering. Specifically, we used the multimodal model Qwen-VL-Max to generate textual descriptions of each LoRA's cover images. These captions were then used to generate new images with both the base model and the base model augmented by the corresponding LoRA. To evaluate whether a LoRA provides meaningful enhancement, we compared the generated images using models such as MPS, HPS, and VQAScore, assessing improvements in both aesthetic quality and text–image alignment. Only LoRAs that achieved consistent improvements across both aspects were retained, resulting in a curated pool of 162 LoRAs.

## C.2 DATASET CONSTRUCTION

Our training dataset is structured as $\mathcal{T} = \{(I_{1,1}, x_{1,1}, (B, A)_1), \cdots, (I_{j,t}, x_{j,t}, (B, A)_j)\}$ where $I_{j,t}$ and $x_{j,t}$ denote the $t$-th cover image of LoRA $j$ and its corresponding textual captions, respectively. Text captions are generated via the Qwen-VL-Max API using the following system prompt:

```
You are a professional image captioner.
Generate a detailed caption according to the image so that another image
    generation model can generate the image via the caption.
The caption should not only describe the content of the image, but also
    include details about its overall style.
Just return the string description, do not return anything else.
```

For the retrieval and fusion experiments' test set, we randomly select 1–3 LoRAs from the candidate pool along with one of their cover images, and generate a joint prompt using the Qwen-VL-Max API. Unlike the training captions, these prompts integrate features from multiple images, thereby simulating fusion scenarios. The system prompt used is:

```
[ROLE]
You are a professional image captioner with expertise in AI model
    evaluation.
[TASK]
Generate a detailed image description (no more 150 words, no line breaks)
    for LoRA model fusion testing.
[INPUT]
Two or three LoRA-generated images with distinct features/styles.
[INSTRUCTION]
1. Identify and merge key elements from both images
2. Prioritize dominant subject (person/object) while integrating
    secondary style
3. Ensure visual logic consistency (e.g., lighting, perspective)
4. Balance style attributes proportionally to input prominence
5. Resolve conflicts with plausible synthesis (e.g., "cyborg cat" to "
    organic-mechanical hybrid")

[FORMAT]
"{Primary subject} {action} {secondary style descriptors} {contextual
    environment}"
stroke texture matching the original painting's impasto technique"
[CONSTRAINTS]
- Maintain semantic coherence
- Avoid abstract metaphors
- Specify style implementation level (subtle/medium/intense)
- Use precise artistic terminology when applicable
```

Although these prompts are informed by LoRA cover images, they differ substantially from the training captions, thereby ensuring diversity while maintaining relevance to the candidate pool—in the sense that associated LoRAs remain represented to some extent within the pool. To further assess generalization, we additionally constructed a test set from DiffusionDB. To enhance prompt diversity and complexity, making them more representative of real-world user intentions, we used the Qwen-Max API to rewrite the original prompts with the following system prompt:

```
You are an expert in text image prompt word polishing.
You need to polish the prompt words input by users so that they are
    suitable for input into the text graph model to generate high-quality
     images. Only the polished prompt word string needs to be returned,
    and no other things need to be returned.
```

## D HUMAN EVALUATION

To evaluate the effectiveness of AutoLoRA in both retrieval and fusion, we conducted user studies with 10 participants. For the retrieval evaluation, we randomly sampled 50 prompts from the Diffu-

Table 4: Human evaluation results for LoRA retrieval and fusion.

| LoRA Retrieval | | LoRA Fusion | |
|---|---|---|---|
| **Method** | **User Preference** | **Method** | **User Preference** |
| FLUX.1 dev | 0.6% | K-LoRA / Direct | 17.2% |
| Text–Image Similarity | 29.4% | | |
| AutoLoRA | **70.0%** | AutoLoRA | **82.8%** |

sionDB dataset and compared three methods: the original FLUX.1 dev model, retrieval followed by fusion using text–image similarity, and retrieval followed by fusion using AutoLoRA. For each case, the top-3 retrieved LoRAs were fused, and users were asked to select the best output based on image quality and semantic alignment. The user preference rates for each method were then recorded.

For the fusion evaluation, we compared AutoLoRA with the strongest baseline methods under two settings. In the case of fusing two LoRAs, we used K-LoRA as the baseline, while for fusing three LoRAs, we adopted direct addition. For each setting, we randomly sampled 50 prompts and asked the same 10 participants to assess outputs in terms of image quality, aesthetics, and the preservation of each LoRA's distinctive features. We report the overall user preference rate across both datasets.

The results in Table 4 consistently show that AutoLoRA is more aligned with human preferences, confirming its superior performance in both LoRA retrieval and fusion tasks.

## E  EFFICIENCY ANALYSIS

To provide a comprehensive efficiency analysis of AutoLoRA, we evaluated both the retrieval and fusion stages in terms of computational cost, GPU memory usage, and inference latency. For the retrieval stage, we computed embeddings for all 162 LoRAs in our collection. The offline embedding computation was performed once and amortized across all queries. At inference time, we measured the cost of retrieving the top-3 LoRAs for a given prompt using cosine similarity search. For the fusion stage, we compared the memory consumption and per-step latency of AutoLoRA against the baseline FLUX model under different numbers of fused LoRAs ($k = 0, 1, 2, 3$). GPU memory usage was recorded using PyTorch's max_memory_allocated() to capture the peak memory footprint during the forward pass. Latency was measured as the average time per diffusion step across 50 steps with batch size set to 1. All experiments were conducted on an AMD ASPEED (rev 52) GPU with 192 GB of memory.

As shown in Table 5, the retrieval stage incurs negligible cost, with top-3 retrieval completed within 0.625 seconds, confirming the efficiency of our retriever. For the fusion stage, memory overhead scales linearly with the number of LoRAs, and latency follows a similar trend: the baseline FLUX runs at 1.02s per step, while AutoLoRA with $k = 3$ increases to 2.11s, roughly doubling inference time. Most of this overhead stems from separately computing forward passes for each LoRA, highlighting a clear direction for future optimization. Overall, AutoLoRA introduces modest and predictable computational overhead while delivering substantial gains in retrieval and fusion quality, making it practical for large-scale deployment.

## F  FAILURE CASES ANALYSIS AND POTENTIALS FOR FUTURE WORKS

Figure 6 presents several representative failure cases observed during our LoRA retrieval experiments, highlighting limitations in current retrieval strategies. In the first example, the input prompt contains keywords such as "nintendo" and "video game," leading to the retrieval of a "pokemon-pixel" LoRA. While there is some semantic overlap between "Pokemon" and the broader concepts of video games or Nintendo, this retrieved LoRA does not align with the user's intended visual outcome. Moreover, due to its pixel-art style, the generated image adopts an unintended aesthetic, significantly deviating from the desired content. Notably, even within the top-3 most similar candidates, a completely irrelevant "home-decoration" LoRA is retrieved, further indicating inaccuracies

Table 5: Efficiency analysis of AutoLoRA. We report offline and online retrieval costs, as well as GPU memory usage and per-step latency during fusion under different numbers of LoRAs ($k = 0, 1, 2, 3$). Results show that AutoLoRA introduces modest overhead while maintaining practical efficiency for deployment.

| Stage | Metric | Setting | Value |
|---|---|---|---|
| Retrieval | Offline embedding computation | 162 LoRAs | 429.265 s |
| | Online Top-k retrieval | Tok-3 LoRAs | 0.625 s |
| Fusion | GPU memory (MB) | k=0 | 35734.33 |
| | | k=1 | 36229.75 (+495.42) |
| | | k=2 | 36547.51 (+813.18) |
| | | k=3 | 37681.76 (+1947.43) |
| | Latency (s/step) | k=0 | 1.02 |
| | | k=1 | 1.39 (+0.37) |
| | | k=2 | 1.90 (+0.88) |
| | | k=3 | 2.11 (+1.09) |

Figure 6: Illustrative retriever failure cases, showing the gap between keyword-level similarity and deeper intent alignment.

in the retrieval process. This suggests that relying solely on text-based semantic similarity may be insufficient for capturing nuanced user intent. The performance of our retrieval system is also constrained by the size and diversity of the LoRA candidate pool. For instance, since no LoRA related to "movie hereditary (2018)" exists in the pool, the system defaults to retrieving semantically proximate yet ultimately irrelevant models, resulting in suboptimal outputs. Another illustrative case involves the prompt "night cityscape," which retrieves a "cyberpunk style" LoRA. This occurs because the training data of this LoRA predominantly consists of neon-lit urban night scenes, creating a strong statistical association between the model and night-time city imagery. While superficially plausible, this reflects a fundamental limitation of the "model-as-semantic" paradigm: the semantics learned by such models are grounded in the distribution of their training data rather than in abstract, human-aligned conceptual understanding. These failure cases underscore two critical challenges: (1) the gap between surface-level keyword matching and deeper intent comprehension, and (2) the dependency of retrieval quality on both the coverage of the LoRA repository and the alignment between model semantics and human perception. They motivate future work toward developing retrieval models that better capture complex instructions and abstract concepts—ideally trained to reflect human preferences rather than merely replicating data-driven co-occurrence patterns.

Figure 7 illustrates some failure cases encountered by our fusion module. For instance, when fusing a "Cyber Style" LoRA with a "Portraits" LoRA, the generated image depicts a person against a cybercity background, rather than a true cyberpunk-style character. This indicates that although the fusion module can preserve individual LoRA characteristics, it still lacks sufficient creative capability—specifically, the ability to combine two LoRAs into something novel and emergent. While our dynamic gating mechanism enables fine-grained control over each LoRA's contribution during generation, at a more fundamental level, this modulation still essentially performs a "weighted" fusion. Although such refined weighting can significantly mitigate conflicts among different Lo-

Figure 7: Representative failure cases of the fusion module, highlighting its limited capacity for creative composition and the suboptimal results when combining multiple semantically conflicting LoRAs.

RAs, achieving genuinely "creative combination" remains a major challenge for the fusion module. Moreover, when merging highly conflicting LoRAs, our fusion gating mechanism may fail entirely, resulting in suppressed contributions from all involved LoRAs or the dominance of only one. For example, when combining "Chinese-comic," "Cinematic-warm-light," and "Ral-dissolve-style," the fusion module exhibits only the features of "Chinese-comic."

## G OTHER ABLATION STUDIES

### G.1 THE IMPACT OF DIFFERENT GATES

We further investigate the impact of different gating mechanisms on the fusion module by conducting an ablation study. During training, we retain only the LoRA-Specific Gate $w_l$ while keeping all other components unchanged. Experiments are conducted on the three-LoRA fusion test set. Results in Table 6 indicate that retaining only the LoRA-Specific Gate yields a slight advantage in LoRA similarity, but leads to a decline in image quality and text–image alignment. This suggests that the Cross-Interaction Gate plays a crucial role in capturing inter-LoRA feature interactions and, through fine-grained regulation, alleviates conflicts among LoRAs during fusion.

Table 6: Ablation on Gating Mechanisms in the Fusion Module.

| Gate | MPS | HPS | VQA | $l_1$-Sim | $l_2$-Sim | $l_3$-Sim |
|---|---|---|---|---|---|---|
| $w_l$ | 18.594 | 0.338 | 0.929 | 0.861 | **0.854** | **0.856** |
| $w_l + w_c$ | **18.611** | **0.339** | **0.931** | **0.862** | **0.854** | 0.854 |

### G.2 IMPACT OF GLOBAL LORA

We investigates the impact of incorporating a constructed Global LoRA into training and inference for LoRA fusion. To this end, we retrain the gated fusion model on 162 LoRA models without Global LoRA integration, while keeping all other configurations fixed. Experiments are conducted on three LoRA fusion datasets, and the results are reported in Table 7. The findings show that incorporating Global LoRA into both training and inference significantly enhances multi-LoRA

fusion performance, with particularly notable gains in the aesthetic quality of generated images. This suggests that Global LoRA provides a shared semantic representation that captures and integrates global information across multiple LoRAs, thereby alleviating conflicts and improving coherence during fusion. Moreover, we observe improvements in individual LoRA similarity, indicating that Global LoRA not only mitigates interference but also enables the fusion module to preserve and emphasize the unique characteristics of each LoRA.

Table 7: The impact of Global LoRA integration on LoRA fusion performance.

| Method | MPS | HPS | VQA | $l_1$-Sim | $l_2$-Sim | $l_3$-Sim |
|---|---|---|---|---|---|---|
| W.o/ Global LoRA | 18.451 | 0.333 | 0.924 | 0.861 | 0.853 | 0.851 |
| W/ Global LoRA | **18.611** | **0.339** | **0.931** | **0.862** | **0.854** | **0.854** |

### G.3 EFFECT OF GLOBAL LoRA RANK

Table 8 reports the impact of varying the rank of the global LoRA on fusion performance. We kept all other parameters fixed and only varied the rank during matrix factorization, evaluating ranks of 2, 4, 8, and 16 when fusing three LoRAs. The quantitative results indicate that the overall performance is not highly sensitive to the choice of rank. However, we observed a tendency for the aesthetic quality of generated images to degrade as the rank increases. A plausible explanation is that higher ranks introduce redundancy: while they capture more fine-grained LoRA features, they also preserve conflicting or noisy components across different LoRAs. This reduces the regularizing effect of low-rank approximation, thereby amplifying inconsistencies in the fused representation and compromising visual quality. Moreover, higher-rank decompositions incur greater computational costs, as they require more parameters and longer training or inference time. Table 9 further summarizes the time required to construct a global LoRA at different ranks. Considering both effectiveness and efficiency, we adopt rank=4 as the default setting for our main experiments.

Table 8: Effect of different decomposition ranks in the global LoRA.

| Rank | MPS | HPS | VQA | $l_1$-Sim | $l_2$-Sim | $l_3$-Sim |
|---|---|---|---|---|---|---|
| Rank=2 | 18.543 | 0.338 | 0.930 | 0.86 | 0.85 | 0.85 |
| Rank=4 | 18.557 | **0.339** | **0.932** | 0.86 | 0.85 | 0.85 |
| Rank=8 | **18.558** | 0.338 | 0.930 | 0.86 | **0.86** | 0.85 |
| Rank=16 | 18.530 | 0.337 | 0.929 | 0.86 | 0.85 | 0.85 |

Table 9: Computation cost of constructing a global LoRA under different decomposition ranks.

| Metric | Rank=2 | Rank=4 | Rank=8 | Rank=16 |
|---|---|---|---|---|
| Time (s) | 0.865 | 1.11 | 1.524 | 2.358 |

### G.4 IMPACT OF TRAINING STRATEGY

To evaluate the effectiveness of our Interference-Resistant training strategy, we compare it against a baseline where the fusion module is trained with only a single LoRA loaded at a time. The training LoRA set remains the same for both methods, and each is trained for 8,000 steps. For evaluation, we construct fusion test sets comprising 300 prompts with 2 LoRAs and 300 prompts with 3 LoRAs, following the procedure described earlier, and report average scores across all metrics.

Results in Table 10 show that while single-LoRA training can preserve the characteristics of individual adapters, it falls short in aesthetic quality and text–image alignment compared to our interference-resistant approach. This is because, when multiple LoRAs are combined, their overlapping or divergent features may superpose uncontrollably, leading to feature entanglement and degraded visual quality (e.g., distorted object shapes, incoherent styles, and mixed artifacts). In

contrast, our strategy enables the fusion module to retain each adapter's beneficial traits while suppressing conflicting contributions, resulting in more harmonious integration, higher aesthetic scores, and stronger alignment.

Table 10: Quantitative comparison between the Interference-Resistant training strategy and the Single-LoRA Training on multi-LoRA fusion tasks

| Training | MPS | HPS | VQA | $l_1$ Sim | $l_2$ Sim | $l_3$ Sim |
|---|---|---|---|---|---|---|
| Single-LoRA Training | 18.449 | 0.338 | 0.926 | 0.857 | 0.852 | **0.860** |
| Interference-Resistant | **18.507** | 0.338 | **0.931** | **0.858** | 0.852 | 0.858 |

## G.5 DIFFERENT LoRA ENCODERS

To demonstrate the necessity of our proposed LoRA Encoder, we compare it against two simple baselines:(1) MLP Encoder. A naive approach is to encode LoRA weights using a simple MLP. However, given the large parameter size of each LoRA layer, directly flattening the $A$ and $B$ matrices and feeding them into a linear layer is infeasible, especially since LoRAs may have different ranks, making the input dimension inconsistent. To address this, we first sum along the rank dimension within each LoRA layer, then flatten and concatenate the resulting $A$ and $B$ matrices. Each concatenated vector is passed through a linear layer, and the outputs from all layers are summed together. Finally, the aggregated representation is fed into a two-layer MLP with a hidden dimension of 768. Despite this simplification, the model still contains 3.68B parameters—an order of magnitude larger than our weight encoder, which only has 112.9M parameters. (2) Fixed-$q_i$. Based on our LoRA Encoder, this baseline initializes each probe vector $q_i$ randomly but keeps them fixed during training, preventing any adaptation.

Table 11 presents the intra-class and inter-class similarity results across six LoRA categories. Compared to the MLP Encoder, our proposed encoder achieves better performance with far fewer parameters. The fixed-$q_i$ method performs worst in terms of intra-class similarity, highlighting the importance of learnable probe vectors. For inter-class similarity, all three methods achieve comparable results (all negative), indicating that LoRAs are well separated in the learned semantic space. The higher intra-class similarity of our method shows that training enables the encoder to capture shared characteristics of LoRAs with similar functions. In particular, learnable $q_i$ vectors contribute significantly by automatically discovering functional features of each LoRA, which brings LoRAs of the same category closer together in the semantic space.

Table 11: Ablation on the LoRA Encoder. Our method yields higher intra-class similarity with fewer parameters while maintaining inter-class separation.

| Metric | Method | Cyber | Ghibli | Mecha | Pixel | Face | 3D | Avg. |
|---|---|---|---|---|---|---|---|---|
| Intra-class($\uparrow$) | MLP | 0.457 | **0.435** | 0.447 | 0.512 | 0.403 | 0.302 | 0.426 |
| | Fixed-$q_i$ | **0.513** | 0.364 | 0.432 | 0.483 | 0.384 | 0.324 | 0.417 |
| | Ours | 0.492 | 0.434 | **0.453** | **0.600** | **0.474** | **0.368** | **0.471** |
| Inter-class($\downarrow$) | MLP | -0.036 | -0.033 | 0.017 | 0.014 | -0.167 | **-0.002** | -0.034 |
| | Fixed-$q_i$ | -0.025 | **-0.051** | 0.019 | **0.009** | **-0.170** | 0.002 | **-0.036** |
| | Ours | **-0.043** | -0.019 | **0.015** | 0.042 | -0.169 | 0.036 | -0.023 |

