# OpenReview forum: "AutoLoRA: Automatic LoRA Retrieval and Fine-Grained Gated Fusion for Text-to-Image Generation"
_ICLR.cc/2026/Conference — ICLR 2026 Conference Withdrawn Submission_

### Official Review · Reviewer_Q52K · 2025-10-30

**Soundness:** 1
**Presentation:** 3
**Contribution:** 3
**Rating:** 2
**Confidence:** 4

**Summary:**

AutoLoRA proposes a framework to address the significant challenge of retrieving and fusing the rapidly growing ecosystem of community-developed LoRAs for diffusion models. The authors introduce a two-stage framework. The first stage consists of a LoRA weight encoding retriever, which is notably designed to encode the model parameters themselves into a semantic space, claiming that conceptually similar LoRA weights cluster together. The second stage employs a gated fusion mechanism, which reportedly overcomes the interference issues common in naive LoRA merging. The authors claim this framework allows for the effective combination of multiple LoRAs (two or more), enhancing generation fidelity and aesthetic quality.

**Strengths:**

1. **Novelty of "Model as Semantic"**: The core idea of encoding the LoRA weights directly, rather than relying solely on textual metadata, is novel and highly promising. The intuition that the model parameters (weights) themselves hold semantic meaning, akin to word embeddings in NLP, is insightful. Using this embedding space for retrieval is a strong and original conceptual contribution.

2. **Methodological Simplicity**: The architecture designed for this weight embedding—projecting LoRA modules as individual tokens and processing them through a Transformer block—is straightforward and elegant.

3. **Grounded Training Strategy**: The use of a CLIP-style contrastive learning objective to align the text semantics (which later serve as user prompts) with the LoRA weight embeddings is a well-reasoned approach that builds upon proven techniques in representation learning.

4. **Intuitive Fusion Mechanism**: The gated fusion method, particularly the use of a global LoRA to stabilize the process and prevent the performance degradation often seen when merging multiple specialized tasks, is an intuitive and practical design choice.

**Weaknesses:**

My recommendation for rejection is primarily driven by severe concerns regarding the paper's experimental soundness, which are detailed below.

### Primary Concern:
- **Potential for Circular Reasoning (Cherry-Picking)**: My most significant concern, leading to the "Poor" soundness score, is the potential for selection bias in the experimental validation. In Appendix C (e.g., Lines 698-701), the authors state that from an initial pool of 1,100 LoRAs, they "retained" a curated pool of 162 based on "consistent improvements across both aspects." This filtering criterion is highly problematic. If the 162 LoRAs were selected because they already produced good results or were known to be compatible, then training and evaluating the system on this pre-filtered set does not validate the method's generalizability. It may only demonstrate that the model works on a subset of LoRAs that were already known to be effective. This fundamentally undermines the central claim that the framework can navigate and manage a large, "in-the-wild" ecosystem of LoRAs.

- **Unusual Efficacy of Contrastive Learning**: Related to the point above, the reported success of the contrastive learning phase is surprising given the small scale of the dataset. CLIP-style training is notoriously data-hungry, often requiring very large batch sizes (e.g., in the tens of thousands) to provide a sufficient number of negative samples for effective learning. The paper reports training on a pool of only 162 LoRAs (which, even when broken down by module, seems small—perhaps ~10k tokens). It is unclear how this relatively small sample size provides enough signal for the contrastive loss to learn a robust and generalizable semantic space. This exceptional result requires a much more detailed explanation and ablation, as it seems to contradict common findings in representation learning. This, combined with the first point, raises significant doubts about the robustness of the trained encoder.

- **Apparent Lack of Proofreading (Potential LLM Artifact)**: I found a concerning artifact in the text that suggests a lack of careful proofreading, possibly stemming from an un-reviewed use of an LLM. In Section 3.1 (Lines 148-149), the text reads, "...the input dimension of the linear layer (corresponds to k in your notation)." The phrase **"your notation"** is highly irregular in academic writing, as it appears to address the reader (or perhaps a co-author/LLM) directly. This calls into question the diligence of the paper's preparation and review process.

### Minor Weaknesses
  1. Data Collection Opacity: The methodology for collecting the initial 1,100 LoRAs is opaque. The provided link seems to lack the necessary metadata to understand how this dataset was assembled or curated.
  2. Incomplete Qualitative Results: The qualitative results presented in Figure 2 are missing the corresponding prompts used for generation. This makes it difficult to properly assess claims of "aesthetic quality," as "detailed" does not automatically mean "aesthetic" or "prompt-aligned."

**Questions:**

I strongly encourage the authors to address the following points in their rebuttal, as they are critical to validating the paper's claims.

### Questions
1. **Clarification on Filtering**: (Relating to Weakness #1) Could the authors please clarify the 1,100-to-162 LoRA filtering process? Specifically, were the "consistent improvements" measured before or after applying the proposed method? How can the authors alleviate the serious concern of selection bias and demonstrate that their method is not just working on a "cherry-picked" dataset?

2. **Contrastive Learning Rationale**: (Relating to Weakness #2) Can the authors provide a stronger rationale or additional experiments (e.g., ablations on dataset size) to justify why the contrastive learning was effective with such a small dataset (162 LoRAs), which seems contrary to common practice?

3. **LLM Specification**: I request a clarification on the specific language models used for different tasks. Section 4 (Experimental Setup) generically mentions using "an LLM," presumably for experimental prompt generation. Appendix C.1 explicitly states the VLM Qwen-VL-Max was used for the data filtering task (generating captions). The appendix also appears to mention the LLM Qwen-Max (e.g., in the context of templates).

  - Could the authors please explicitly confirm which model was used for the "LLM" task described in Section 4? Was it Qwen-Max, the text-generation capability of Qwen-VL-Max, or another model entirely? A clear distinction is needed.

4. **Prompt Diversity Metric**: Is there any quantitative metric to demonstrate that the LLM-generated prompts (Sec 4) are indeed more diverse or complex than, for example, the original showcase prompts from the LoRA pages?

5. **Methodological Choice (VLM vs. Image)**: The authors used a VLM to generate text descriptions from showcase images for the contrastive alignment. Why was this indirect approach chosen over a more direct alignment, such as a Weight-to-Image contrastive loss? The showcase images themselves would seem to be the most representative "ground truth" for a LoRA's concept, with VLM-generated text being a secondary, and potentially noisy, representation.

6. **Human Evaluation Details**: Regarding the human evaluation study (which seems like a crucial result):

- Why was this study placed in the Appendix rather than the main paper?

- The study involved 10 human participants. Was there any form of IRB (Institutional Review Board) approval or ethical consideration process?

- Could the authors provide details on the evaluation UI and the exact instructions or questions given to the evaluators?

7. **Scalability**: Given that the model was trained on a curated set and the failure cases show limitations in retrieval, what are the authors' thoughts on the scalability of this approach? How well would it realistically perform on a truly open-domain, uncurated set of new LoRAs from the community?

### Concluding Remarks
The core idea of **model as semantic** for LoRA weights is highly interesting and a valuable research direction. However, the current submission is critically undermined by severe concerns regarding its experimental soundness, primarily the potential for selection bias in the dataset. While I cannot recommend acceptance in its current state, I am willing to reconsider my evaluation if the authors can provide convincing clarifications and evidence in their rebuttal that fully resolve these fundamental validation issues.

### Ethics and LLM Disclosure
I have used an LLM to assist with improving the grammar and clarity of this review. The content, analysis, and final judgments are my own. This review has been written in accordance with the ICLR Code of Ethics.

---

### Official Review · Reviewer_4vu2 · 2025-10-30

**Soundness:** 2
**Presentation:** 3
**Contribution:** 3
**Rating:** 6
**Confidence:** 3

**Summary:**

The paper addresses the problem of leveraging the vast ecosystem of low-rank adaptation modules (LoRAs) for improving text-to-image generation. The authors introduce a retrieval mechanism that embeds LoRA weight matrices into a shared semantic space aligned with text prompts, along with a fine-grained gated fusion mechanism that dynamically computes context-specific fusion weights across network layers and diffusion timesteps, allowing for the combination of multiple LoRAs for generation.

**Strengths:**

1. The paper introduces a strategy for encoding LoRA weights to enable effective retrieval.
2. It proposes a novel gated fusion mechanism for combining multiple LoRAs, which does not require training on specific LoRAs and scales independently of their number.
3. The authors conduct extensive experiments and ablation studies to demonstrate the contribution of each component in their method.

**Weaknesses:**

1. The paper includes a limited number of visual examples; additional qualitative results would help to better assess output quality.
2. Some proofreading is needed, especially in section 3. For instance, line 149 contains: “.. the input dimension of the linear layer (corresponds to k in *your* notation)”
3. It is unclear whether the retrieved LoRAs are indeed the most relevant ones, and whether the text descriptions of generated images sufficiently capture the characteristics of each LoRA, see q.1-2 below.
4. The proposed method does not necessarily outperform prior work or the “original” baseline, and improvements appear marginal in some cases (e.g. tables 2-3). More visual examples with relevant prompts could help better demonstrate its advantages.

**Questions:**

1. What was the prompt to the VLM to convert each LoRA rendering into text? did you ask for style specifically? If the captions emphasize objects rather than style, retrieval may favor LoRAs depicting similar content rather than the intended visual style.
2. Following the above - did you evaluate the quality of retrieved LoRAs? e.g., what percentage of retrieved LoRAs are relevant to the target prompt, and specifically to the desired style?
3. The method assumes that LoRAs are consistent, of high quality, and that their published examples and representative of them. Did you verify that these assumptions hold for your selected subset?
4. What were the input prompts for Figure 2? It will be easier to assess which image is better with the required prompts..

**Details Of Ethics Concerns:**

No concerns.

---

### Official Review · Reviewer_xF7H · 2025-10-31

**Soundness:** 3
**Presentation:** 2
**Contribution:** 2
**Rating:** 2
**Confidence:** 4

**Summary:**

The paper presents AutoLoRA, a framework designed to enhance text-to-image generation by systematically leveraging the fragmented ecosystem of community-contributed LoRA adapters. It tackles three challenges—sparse metadata, zero-shot adaptation, and multi-LoRA fusion—through two ways: a weight encoding–based LoRA retriever, which learns to align LoRA parameters with text prompts in a shared embedding space using contrastive learning, and a fine-grained gated fusion mechanism that dynamically integrates multiple LoRAs at the layer and timestep levels. This allows AutoLoRA to retrieve relevant adapters without access to training data and fuse up to three distinct LoRAs without catastrophic interference. Experiments on both synthetic and real-world prompts demonstrate improvements in image fidelity, aesthetic quality, and text-image alignment compared to strong baselines.

**Strengths:**

The paper addresses a timely and practical challenge in the use of community-generated LoRA adapters for text-to-image generation, proposing a system that operates without requiring access to training data or metadata. Its main contribution lies in introducing a weight-based retrieval mechanism that encodes LoRA parameters into a shared embedding space with textual prompts using contrastive learning. This design is original in attempting to interpret LoRA weights semantically, without relying on input-output examples. The gated fusion module adds another dimension of novelty, enabling layer-wise, timestep-specific integration of multiple LoRAs—a refinement over prior fixed-weight or MoE-based methods. The authors also conduct controlled experiments to evaluate performance under both synthetic and real prompt settings.

**Weaknesses:**

While the paper presents an interesting attempt to formalize LoRA retrieval and fusion, several critical weaknesses limit its credibility and contribution. First, the retriever training setup appears fundamentally underpowered: the model is trained on only 162 LoRA modules, an extremely small dataset for contrastive learning of this type. Given that CLIP-style embeddings require hundreds of thousands of diverse examples to generalize, it is doubtful that meaningful cross-modal alignment could emerge from such limited data. The absence of training diagnostics, ablations on data scale, or released checkpoints further undermines reproducibility and raises concerns about the reliability of the reported results. Second, the fusion mechanism offers little novelty, as the proposed Fine-Grained Gated Fusion closely mirrors existing gating or mixture-of-experts methods (e.g., MoLE, DARE) and does not provide a clear conceptual or empirical advance. Finally, the experimental comparison is incomplete, omitting stronger recent baselines such as LoRA Composer, ZipLoRA, and K-LoRA, which already perform dynamic or adaptive fusion. Without these comparisons, the paper’s claims of superior performance and scalability remain unconvincing.

**Questions:**

1. Please provide justification for how a retriever trained on only 162 LoRAs can produce meaningful contrastive embeddings, and clarify its robustness to data scale.

2. Please release the retriever model and training details to enable independent verification of the reported results.

3. Clarify the technical differences between the proposed Gated Fusion and prior methods such as MoLE or DARE.

4. Include stronger recent baselines (e.g., LoRA Composer, ZipLoRA) in the experimental comparison for a fair evaluation.

5. Provide evidence of scalability by evaluating the method with more than three LoRAs fused simultaneously.

---

### Official Review · Reviewer_Cn3A · 2025-10-31

**Soundness:** 2
**Presentation:** 2
**Contribution:** 2
**Rating:** 4
**Confidence:** 4

**Summary:**

The paper proposes a framework that unifies community-developed LoRA adapters through semantic retrieval and dynamic fusion, effectively functioning as an ecosystem integrator.

**Strengths:**

- **Motivation is reasonable**:
   Clear formulation of a weight‑probe encoder for data‑free LoRA semantics and cross‑interaction gating with layerwise scaling; ablations show the gate helps and Global‑LoRA improves fusion.

- **Empirical results cover multiple aspects**: Multi-aspect covered evaluations on synthetic prompts, rewritten DiffusionDB prompts, object–style fusion, and random multi‑LoRA fusion with consistent gains.

- **Clarity of Figures and Diagrams**: The embedding clusters by theme and interpretable gate dynamics over timesteps and diagrams are clear.

**Weaknesses:**

- **Motivation is Good, but experimental setup is naive**: The experimental setup for generation model like this is too naive and hard to delivery useful signal for community to deploy or explore this direction. Using simple concept / styling LoRAs are not useful.

- **Experimental Setup Fairness**: LoRAs are pre‑filtered to those that improve MPS/HPS/VQA, and many prompts are synthetic/rewritten, reducing ecological validity.

- **Compare to Model Souping**:  There are previous works in the direction of model souping and also some existing methods in the community to employ multiple LoRAs (especially LoRAs for multi-purpose, distillation / concept stylization / bi-directional to auto-regressive / higher resolution ). The comparison and study in these setup are required.

**Questions:**

I think the major concern for this paper is the experimental setup for mixture multiple LORA with simple concept or style LoRAs are too naive. There is a need to setup more practical use of multi-domain LoRAs and experiment the combination: e.g., LoRA A for efficiency (4-step distillation), LoRA B for stylization, LoRA C for complex prompting.

---

### Note · Authors · 2025-11-14

I have read and agree with the venue's withdrawal policy on behalf of myself and my co-authors.